# The Impact of Lens Epithelium-Derived Growth Factor p75 Dimerization on Its Tethering Function

**DOI:** 10.3390/cells13030227

**Published:** 2024-01-25

**Authors:** Tine Brouns, Vanda Lux, Siska Van Belle, Frauke Christ, Václav Veverka, Zeger Debyser

**Affiliations:** 1Laboratory for Molecular Virology and Gene Therapy, Department of Pharmaceutical and Pharmacological Sciences, KU Leuven, 3000 Leuven, Flanders, Belgium; tine.brouns@kuleuven.be (T.B.); siska.vanbelle@kuleuven.be (S.V.B.); frauke.christ@kuleuven.be (F.C.); 2Institute of Organic Chemistry and Biochemistry of the Czech Academy of Sciences, 16000 Prague, Czech Republic; vanda.lux@uochb.cas.cz (V.L.); vaclav.veverka@uochb.cas.cz (V.V.); 3Department of Cell Biology, Faculty of Science, Charles University, 12800 Prague, Czech Republic

**Keywords:** DNA-binding protein, protein–DNA interaction, protein–protein interaction, chromatin structure, protein dynamic, LEDGF/p75, DNA-induced protein binding

## Abstract

The transcriptional co-activator lens epithelium-derived growth factor/p75 (LEDGF/p75) plays an important role in the biology of the cell and in several human diseases, including MLL-rearranged acute leukemia, autoimmunity, and HIV-1 infection. In both health and disease, LEDGF/p75 functions as a chromatin tether that interacts with proteins such as MLL1 and HIV-1 integrase via its integrase-binding domain (IBD) and with chromatin through its N-terminal PWWP domain. Recently, dimerization of LEDGF/p75 was shown, mediated by a network of electrostatic contacts between amino acids from the IBD and the C-terminal α_6_-helix. Here, we investigated the functional impact of LEDGF/p75 variants on the dimerization using biochemical and cellular interaction assays. The data demonstrate that the C-terminal α_6_-helix folds back in cis on the IBD of monomeric LEDGF/p75. We discovered that the presence of DNA stimulates LEDGF/p75 dimerization. LEDGF/p75 dimerization enhances binding to MLL1 but not to HIV-1 integrase, a finding that was observed in vitro and validated in cell culture. Whereas HIV-1 replication was not dependent on LEDGF/p75 dimerization, colony formation of MLLr-dependent human leukemic THP-1 cells was. In conclusion, our data indicate that intricate changes in the quaternary structure of LEDGF/p75 modulate its tethering function.

## 1. Introduction

Lens epithelium-derived growth factor (LEDGF/p75) is a transcriptional co-activator [1] and cellular stress survival factor [2] with an essential role in development [3] and physiological processes such as DNA repair and RNA transcription [4,5,6]. It has been investigated in much detail due to its role in several diseases such as HIV-1 infection and mixed-lineage leukemia-rearranged (MLL-r) leukemia [7,8]. The integrase-binding domain (IBD) of LEDGF/p75 interacts with the catalytic core domain of integrase (IN), whereby it tethers and targets HIV-1 viral DNA to the chromatin for integration into transcriptionally active regions [7,9,10,11]. Next, LEDGF/p75 was found to play a role in leukemogenesis through its interaction with oncogenic MLL-fusion proteins. Again, MLL1-dependent transcription and leukemic transformation depends on LEDGF/p75 tethering and activation of specific HOX genes [8,12]. Additionally, roles for LEDGF/p75 have been reported in multiple cancers, like breast cancer [13], cervical cancer [14], and human prostate cancer [15,16].

LEDGF/p75 contains several well-defined domains and important interaction motifs (Figure 1a). The N-terminal domain (aa 1–325) contains a PWWP domain (aa 1–93) that functions as a chromatin interaction domain [17,18,19,20], followed by a positively charged region (CR1). C-terminally from the CR1, a nuclear localization signal (NLS) is present (aa 146–156), regulating the nuclear import of LEDGF/p75, and two AT-hook like motifs (aa 178–183 and 193–197), mediating nonspecific DNA binding [21]. Two other charged regions (CR2 and 3) (within the region of aa 208–345) influence the AT-hook function [22], and form the supercoiled DNA-recognition domain (SRD) (aa 200–336) that might recognize active transcription units [23]. Recently, an additional DNA-binding region was identified in the intrinsically disordered domain [19]. The N-terminal part of the protein is shared with the alternative splice variant, LEDGF/p52, except for the last eight C-terminal amino acid residues. Notwithstanding the similarity between LEDGF/p52 and the N-terminal domain of LEDGF/p75, p52 has the strongest transcription activation potential [1]. The C-terminal domain of the LEDGF/p75 protein (aa 326–530) contains the integrase-binding domain (IBD) (aa 345–426), which can interact with HIV-1 integrase (HIV-1 IN), hence its name. This domain comprises ±80 amino acid residues that are both necessary and sufficient for binding to HIV-1 IN [7]. The IBD was later described as the common binding region for all known cellular binding partners of LEDGF/p75 [24], harboring an IBD-binding motif. This motif represents a short linear motif (SLiM), which is an evolutionarily conserved unstructured protein module that adopts a characteristic structure upon binding its target site [25]. The conserved region of this SLiM is within the cellular binding partners of LEDGF/p75 extended with an additional upstream motif, characterized by an alpha-helical propensity and a conserved phenylalanine [26]. In addition, this region serves as an important regulatory switch, which can vary its charge density via phosphorylation of serine and threonine residues present in the SLiM increasing or decreasing the affinity of a TIM for the TND [26].

Regulation of LEDGF/p75 binding to its interaction partners may also be affected by its dimerization [27]. Recently, the biochemical and biophysical properties of LEDGF/p75 dimers were reported, describing the IBD domain-swapped dimer conformation, stabilized by a network of electrostatic contacts established between amino acids from the IBD and the α_6_-helix in the C-terminal domain of the protein (Figure 1b, [28]). The part behind the IBD, from residue 426 to 530, is called the C-term throughout this manuscript.

**Figure 1 cells-13-00227-f001:**
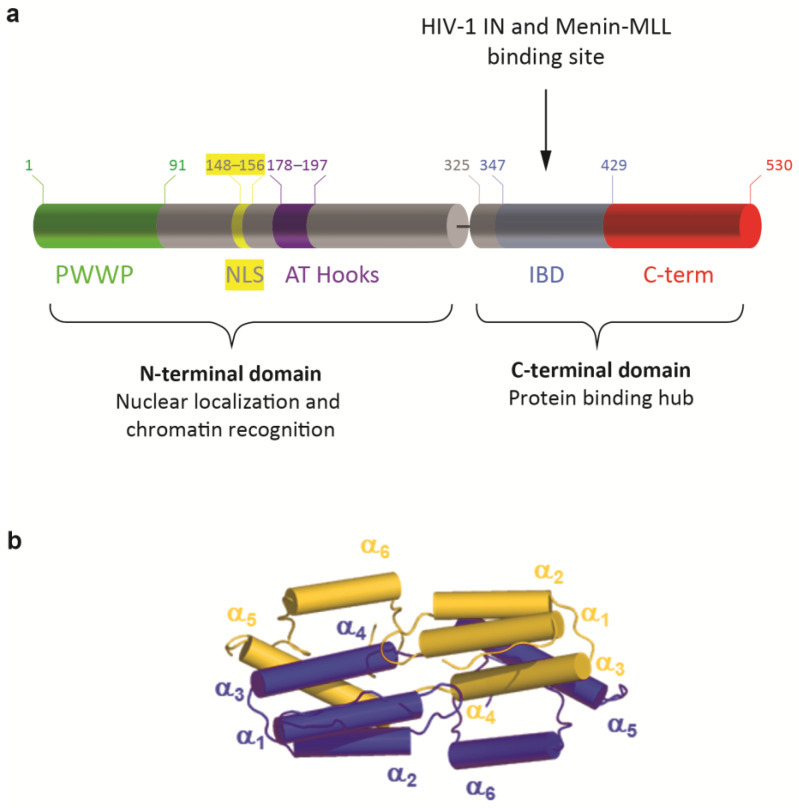
Domain structures of LEDGF/p75. (**a**) At the N-terminal end, LEDGF/p75 contains a Pro-Trp-Trp-Pro (PWWP) domain (aa 1–93), followed by charged regions, including the nuclear localization signal (NLS) and two AT-Hook like domains, forming the N-terminal domain (aa 1–325). The C-terminal domain (aa 325–530) contains an integrase-binding domain (IBD) and a C-terminal end, called C-term throughout this manuscript (aa 429–530). (**b**) Cartoon representation of LEDGF_325–467_ IBD dimerization as recently described by Lux et al. [28]. Within the integrase-binding domain, domain swapping of α_6_-helix, with additionally electrostatic “stapling” of the α_6_-helix formed in the intrinsically disordered C-terminal region, is the foundation for dimeric assembly.

In this study, we investigated the biological relevance of dimerization at nanomolar concentrations. We first tested LEDGF/p75 and its mutants in AlphaScreen, a protein–protein interaction assay. The impact of dimerization was further investigated in two cellular models by analyzing the mutants in a HIV-1 cellular replication assay and in a colony formation assay to investigate MLL1 leukemogenesis. Different aspects of an intricate mechanism to regulate the tethering function by quaternary structure alterations were identified. We discovered that the AT-hook like regions of LEDGF/p75 mediate DNA-induced dimerization, which is required for the interaction with MLL1, but not with HIV-1 integrase. Next, we observed that the C-term of LEDGF/p75 not only binds to the IBD of another LEDGF/p75 molecule to form a swapped dimer, but that the C-term of the protein can fold back in *cis* on its own IBD, at low protein concentrations.

## 2. Materials and Methods

### 2.1. Plasmids

Flag-LEDGF_dDNA_ was previously described in [29]. The pCp-NAT-3xFlag-LEDGF WT construct was used to modify full-length Flag-tagged proteins, the pET-His-TEV-LEDGF_345–530_ plasmid was used to modify the truncated C-terminal LEDGF mutants, and the pMBP-Δ325 [7] plasmid was used to generate MBP-LEDGF_325–467_ and MBP-LEDGF_325–426_. The Site-directed Ligase-Independent Mutagenesis [30] protocol was used to generate the correct constructs with corresponding primer set, given in Appendix A. All constructs were verified by DNA sequencing. GST-LEDGF/p75 was expressed from pGEX-p75, a plasmid kindly provided by Mamuka Kvaratskhelia (The Ohio State University).

### 2.2. Protein Expression and Purification

All LEDGF constructs were transformed in *E. coli BL21*(DE3)pLysS (Oneshot, Invitrogen by Thermo Fisher Scientific, Waltham, MA, USA) competent cells by heat shock and grown in Lysogeny broth (LB, Sigma, St. Louis, MO, USA) supplemented with 100 μg/mL ampicillin (Sigma, St. Louis, MO, USA), and 2 g/L glucose (Sgima, St. Louis, MO, USA) for MBP-tagged proteins. Bacterial cultures were grown at 37 °C (flag-tag) or 30 °C in case of His_6_-tagged proteins. After an OD_600_ of 0.6 was reached, protein expression was induced with 0.5 mM isopropyl-β-D-thiogalactopyranoside (IPTG, Sigma, St. Louis, MO, USA). Proteins were induced for four hours of induction at 30 °C for flag-tagged proteins, four hours of induction at 37 °C for GST-tagged proteins, and overnight induction at 18 °C for His_6_-tagged proteins. After incubation, cultures were harvested by centrifugation for 10 min at 5000 rpm using the Fiberlite™ F12-6 × 500 LEX Fixed Angle Rotor (Thermo Fisher Scientific, Waltham, MA, USA). Pellets were washed in cold STE buffer containing 10 mM Tris-HCl (Sigma, St. Louis, MO, USA), pH 7.3, 100 mM NaCl (Sigma, St. Louis, MO, USA), and 0.1 mM EDTA (Invitrogen by Thermo Fisher Scientific, Waltham, MA, USA), and centrifuged as described before. Pellets were stored at −20 °C.

Cells were resuspended in lysis buffer (50 mM Tris-HCl(Sigma, St. Louis, MO, USA), pH 7.4, 100 mM NaCl(Sigma, St. Louis, MO, USA), 1 mM dithiothreitol (DTT; Sigma, St. Louis, MO, USA), 0.1 μg/mL DNase (Thermo Fisher Scientific, Waltham, MA, USA) and protease inhibitor (cOmplete, EDTA-free, Roche, Basel, Switzerland) and lysed by sonication using the MSE 150 Watt Ultrasonic Disintegrator. The lysate was cleared by 30 min of centrifugation at 15,000 rpm using the Fiberlite™ F21 × 50y Fixed Angle Rotor (Thermo Fisher Scientific, Waltham, MA, USA) and 4 °C.

For Flag-tagged proteins, the cleared lysate was first filtered through a Millex-GS Syringe Filter Unit of 0.22 µm (Millipore, Burlington, MA, USA) and subsequently loaded on an Äkta Purifier (GE Healthcare, Chicage, IL, USA), equilibrated with running buffer (50 mM Tris-HCl, pH 7.4, 150 mM NaCl and 1 mM DTT; all reagents from Sigma, St. Louis, MO, USA) and connected with a 5 mL HiTrap heparin column (GE Healthcare, Chicage, IL, USA). The protein was eluted by a NaCl gradient in 50 mM Tris, pH 7.5 (reagents from Sigma, St. Louis, MO, USA). Elution fractions were run on an in-house-made SDS-PAGE gel to select the correct LEDGF fraction, which was pooled and concentrated by use of an Amicon^®^ Ultra Centrifugal Filter, until volumes of maximum 1 mL. The sample was further purified on a Superdex 75 10/300 GL column (GE Healthcare, Chicago, IL, USA). At last, fractions were analyzed for protein content with an in-house-made SDS-PAGE gel and stained with Coomassie blue.

GST-tagged proteins were purified as described before [24].

For His_6_-tagged proteins, 20 mM imidazole (Acros Organics, Geel, Belgium) was added to the lysis buffer. Purification by affinity chromatography was carried out on a Ni-NTA-resin (Thermo Fisher Scientific, Waltham, MA, USA). The resin was washed in 50 mM Tris-HCl (Sigma, St. Louis, MO, USA), pH 7.5, 150 mM NaCl (Sigma, St. Louis, MO, USA), and 20 mM imidazole (Acros Organics, Geel, Belgium), and His_6_-tagged proteins were eluted in wash buffer with increased imidazole concentration (230 mM). The protein was additionally purified over a HiTrap Heparin HP resin (GE Healthcare, Chicage, IL, USA) equilibrated with 50 mM Tris-HCl, pH 7.5, and 150 mM NaCl (all reagents from Sigma, St. Louis, MO, USA). The protein was eluted by a NaCl gradient in 50 mM Tris, pH 7.5, using the FPLC Äkta Purifier system (GE Healthcare, Chicage, IL, USA).

MBP-tagged proteins were purified by applying the supernatant onto 2 mL of previously washed (50 mM Tris-HCl (Sigma, St. Louis, MO, USA), pH 7.2, 500 mM NaCl (Sigma, St. Louis, MO, USA), 1 mM EDTA (Invitrogen by Thermo Fisher Scientific, Waltham, MA, USA) amylose resin (NEB, Ipswich, MA, USA). Here, 10 mM maltose was supplemented to the wash buffer for elution. Peak fractions were pooled and maltose was removed by dialysis against 50 mM Tris-HCl (Sigma, St. Louis, MO, USA), pH 7.2, 500 mM NaCl (Sigma, St. Louis, MO, USA), and 10% (*v*/*v*) glycerol (Acros Organics, Geel, Belgium).

HIV-1 integrase was expressed from pKB-IN6H and grown in similar conditions as LEDGF constructs, except for the addition of 25 mg/L chloramphenicol (Sigma, St. Louis, MO, USA) in LB medium (Sigma, St. Louis, MO, USA). Protein expression was induced after addition of 0.5 mM IPTG (Sigma, St. Louis, MO, USA) at an OD_600_ of 0.8. The bacteria were grown for 3 h at 29 °C and harvested by spinning down for 30 min at 5000 rpm using the Fiberlite™ F12-6 × 500 LEX Fixed Angle Rotor (Thermo Fisher Scientific Waltham, MA, USA). Pellets were washed in STE buffer, and stored at −20 °C. The cells were lysed in buffer containing 25 mM HEPES (Sigma, St. Louis, MO, USA), pH 7.4, 1 M NaCl (Sigma, St. Louis, MO, USA), 1 mM MgCl_2_ (Sigma, St. Louis, MO, USA), 7.5 mM 3-[(3-cholamidopropyl)dimethylammonio]-1-propanesulfonate (CHAPS; VWR, Radnor, PA, USA), and 3 mM dithiothreitol (DTT; Sigma, St. Louis, MO, USA)), supplemented with 0.1 μg/mL DNase (Thermo Fisher Scientific, Waltham, MA, USA), protease inhibitor (cOmplete, EDTA-free, Roche, Basel, Switzerland), and 20 mM imidazole (Acros Organics, Geel, Belgium), and lysed by sonication (MSE 150 Watt Ultrasonic Disintegrator). The lysate was cleared by 30 min of centrifugation at 15 000 rpm, using the Fiberlite™ F21 × 50y Fixed Angle Rotor (Thermo Fisher Scientific Waltham, MA, USA), and 4 °C, after which the supernatant was applied onto 2 mL of previously washed (25 mM HEPES (Sigma, St. Louis, MO, USA), pH 7.4, 1 M NaCl (Sigma, St. Louis, MO, USA), 1 mM MgCl_2_ (Sigma, St. Louis, MO, USA), 7.5 mM CHAPS (VWR, Radnor, PA, USA), and 3 mM DTT (Sigma, St. Louis, MO, USA) Ni-NTA-agarose (Qiagen, Hilden, Germany). The protein was eluted with wash buffer supplemented with 180 mM imidazole. Pooled peak fractions were cleared from imidazole by dialysis against wash buffer containing 10% glycerol (Acros Organics, Geel, Belgium).

MLL1_1–160_-GST was expressed from the pET-20b MLL1_1–160_-GST plasmid and grown similarly as LEDGF constructs. After 5 h of induction, cells were harvested by centrifugating for 10 min at 5000 rpm using the Fiberlite™ F12-6 × 500 LEX Fixed Angle Rotor (Thermo Fisher Scientific, Waltham, MA, USA). After washing the pellet in STE buffer, the pellet was stored at −20 °C. Glutathione Sepharose-4 Fast Flow (GE Healthcare, Chicago, IL, USA) was used to selectively purify the GST-tagged protein after the cells were sonicated in buffer containing 50 mM Tris-HCl (Sigma, St. Louis, MO, USA) pH 7.3, 150 mM NaCl (Sigma, St. Louis, MO, USA), 1 mM DTT (Sigma, St. Louis, MO, USA), 0.1 μg/mL DNase (Thermo Fisher Scientific, Waltham, MA, USA) and protease inhibitor (cOmplete, EDTA-free, Roche, Basel, Switzerland). The resin was equilibrated with wash buffer (50 mM Tris-HCl pH 7.5, 150 mM NaCl; all reagents from Sigma, St. Louis, MO, USA). Proteins were eluted in wash buffer with 25 mM glutathione (Acros Organics, Geel, Belgium). The protein was further purified over a HiTrap Heparin HP resin (GE Healthcare, Chicago, IL, USA) equilibrated with 50 mM Tris-HCl pH 7.5, 150 mM NaCl, and 1 mM DTT (all reagents from Sigma, St. Louis, MO, USA). The protein was eluted by a NaCl gradient in 50 mM Tris pH 7.5 using the FPLC Äkta Purifier system (GE Healthcare, Chicago, IL, USA).

Elution fractions were analyzed for protein content on SDS-PAGE, followed by a Coomassie (Brilliant blue G 250, Merck, Darmstadt, Germany) staining. All selected protein fractions were pooled, and if not dialyzed, supplemented with 10% (*v*/*v*) glycerol (Acros Organics, Geel, Belgium) before aliquoting and snap-freezing. Protein samples were stored at −80 °C. Coomassie stains of the used proteins are available in supporting Figure 1. The amount of DNA was measured using the Qubit 4 (Thermo Fisher Scientific, Waltham, MA, USA) and the 1× Qubit dsDNA HS assay kit (Thermo Fisher Scientific, Waltham, MA, USA).

### 2.3. AlphaScreen Assay

Protein–protein interactions were studied by a bead-based, nonradioactive Amplified Luminescent Proximity Homogeneous Assay Screen (AlphaScreen, Perkin Elmer, Mechelen, Belgium) as described in [28]. In case DNA was digested, 50U of Micrococcal Nuclease (Thermo Fisher Scientific, Waltham, MA, USA) was added to the AlphaScreen buffer, supplemented with 5 mM of CaCl_2_ (Sigma, St. Louis, MO, USA).

### 2.4. Generating Stable Cell Lines

All cells were kept in Dulbecco’s modified Eagle’s medium (DMEM; GIBCO-BRL, Merelbeke, Belgium) supplemented with 5% *v*/*v* heat-inactivated fetal bovine serum (FBS; Gibco, Invitrogen, Waltham, MA, USA) and 0.005% *w*/*v* gentamicin (GIBCO, Waltham, MA, USA). To maintain cell line stability after viral transduction, DMEM medium was supplemented with 0.05% *w*/*v* geneticin (Gibco, Invitrogen, Waltham, MA, USA) for HeLa P4 cells and 0.01% *w*/*v* Zeocin (Life Technologies, Ghent, Belgium) or 0003% *w*/*v* blasticidin (Invitrogen, Waltham, MA, USA), depending on the viral vector resistance cassette. SIV-based viral vectors were prepared as described earlier [31]. Stable LEDGF/p75 KD cells (HeLa P4 LEDGF/p75-depleted cells) were generated as described earlier [7] and selected by zeocin. All LEDGF/p75 mutant expression constructs were cloned into the pGAE backbone and cloning steps were sequence verified. To make the overexpressed stable cell lines, 20,000 HeLa P4 (LEDGF KD) cells were seeded in a 96-well plate and transduced with the corresponding lentiviral vector. After 72 h, selection was initiated by reseeding the cells in a 24-well plate in with the presence of 0.0003% *w*/*v* blasticidin (Invitrogen, Waltham, MA, USA).

THP1 human MLL1-AF9+ AML cells (kind gift of Prof. J. Schwaller, Laboratory of childhood leukemia, Switzerland) were maintained in Roswell Park Memorial Institutes medium (RPMI 1640, Gibco-BRL, Merelbeke, Belgium) supplemented with 10% (*v*/*v*) heat-inactivated fetal bovine serum (FBS; Gibco, Invitrogen, Waltham, MA, USA) and 50 µg/mL gentamycin (Gibco, Invitrogen, Waltham, MA, USA). First, cells were transduced with the lentiviral vector expressing the corresponding mutant. As a control for blasticidin selection, mock cell lines were generated with pGAE-blasti-miR-Fluc. Subsequently, these cells were transduced with the same vector as described above to make LEDGF/p75 KD, and selected with RPMI 1640 medium containing 0.01% *w*/*v* zeocin.

### 2.5. Western Blot

Protein samples were separated by SDS-PAGE and electroblotted onto PVDF membranes (Bio-Rad Laboratories, Hercules, CA, USA). After transfer, membranes were blocked with 5% (*w*/*v*) milk powder in PBS-T (0.1% Tween 20 in PBS). Protein detection was carried out using specific antibodies against LEDGF/p75 C-term (Bethyl Laboratories, Montgomery, TX, USA, A300-848A, 1/500) or LEDGF/p75 PWWP (Abcam, Cambridge, UK, ab177159, 1/1000), Flag (Sigma, St. Louis, MO, USA, F7425, 1/3000), β-tubulin (Sigma, St. Louis, MO, USA, T4026-2, 1/5000) or GAPDH (Abcam, Cambridge, UK, ab9485, 1/1000). Protein bands were visualized by chemiluminescence (ECL+; GE Healthcare, Chicago, IL, USA) and horseradish peroxidase (HRP)-conjugated secondary antibodies.

### 2.6. Quantitative RT-PCR

Using the Aurum Total RNA mini kit (Bio-Rad Laboratories, Hercules, CA, USA), RNA was isolated from the cells. The RNA concentration was measured using the absorbance at 260 nm, and equal amounts of RNA were reverse-transcribed to cDNA with the High Capacity cDNA Reverse Transcription Kit (Thermo Fisher Scientific, Waltham, MA, USA). For each q-PCR reaction, 12.5 µL IQ Supermix (Bio-Rad Laboratories, Hercules, CA, USA) or LightCycler^®^ 480 SYBR Green I Master (Roche, Basel, Switzerland) was added to 250 nM forward and reverse primers and 250 nM probe (if indicated) in a final volume of 20 µL. cDNA was denaturated for 10 min at 95 °C, followed by 50 repeats of 10 s denaturation at 95 °C and 30 s elongation at 55 °C. Detection and analysis were performed in the LightCycler 480 (Roche, Basel, Switzerland). The following primer/probe sets were used: LEDGF/p75_d325_ sense: 5′-GAA CTT GCT TCA CTT CAG GTC ACA-3′ and LEDGF/p75_d325_ antisense: 5′-TCG CCG TAT TTT CAG TGT AGT-3′ and probe: 5′-FAM-TGC AAC AAG CTC AGA AAC ACA CAG AGA TGA -TAMRA-3′ and β-actin sense: 5′-CAC TGA GCG AGG CTA CAG CTT-3′ and β-actin antisense: 5′-TTG ATG TCG CGC ACG ATT T-3′ and probe 5′-HEX-ACC ACC ACG GCC GAG CGG -TAMRA-3′.

### 2.7. HIV-1 Infection

HIV-1 infection of HeLa P4 cells was performed as described before [7]. A total of 3 × 10^4^ cells per well were infected with HIV(NL4.3) in a6 well plate for 6 h. After 6 h, the virus was washed away with PBS (Invitrogen, Waltham, MA, USA) and replaced with 5 mL of medium. Replication of HIV-1 was monitored by quantifying p24 antigen in the supernatant via ELISA (Alliance HIV-1 p24 ELISA kit; PerkinElmer, Mechelen, Belgium).

### 2.8. In Vitro Clonic Growth Assay

For colony-forming unit (CFU) assays, 2000 cells were plated in human methylcellulose (Methocult H4230, Stemcell technologies, Vancouver, BC, Canada) in the presence of appropriate antibiotic selection. After 12 to14 days, the number of colonies was counted.

## 3. Results

### 3.1. Domains of LEDGF/p75 Involved in Dimerization

Scanning force microscopy studies revealed the presence of LEDGF/p75 monomers and LEDGF/p75 dimers, and reported that the LEDGF/p75 conformation affected DNA binding [27]. More recently, Lux V. et al. validated the presence of LEDGF/p75 as both monomer and dimer using size exclusion chromatography, and determined the minimal dimerization domain of the C-terminal domain to amino acids 345–467 by NMR [28]. Here, to probe the (i) homodimerization of LEDGF/p75, (ii) its interaction with DNA, and (iii) the interaction with binding partners, including MLL1 and HIV-1 integrase (IN), we used an in vitro bead-based biomolecular interaction assay: AlphaScreen (Amplified Luminescent Proximity Homogenous Assay). The direct interaction of two differentially tagged LEDGF/p75 molecules (Flag- and GST-tagged) was demonstrated by performing a cross-titration with varying concentrations of each protein (Figure 2a). This cross-titration experiment defined the optimal concentrations (35 nM–350 nM) and ratios for subsequent dimerization assays. Different truncation mutants were generated to pinpoint the dimerization domain (Figure 2b). In line with the previously published NMR-based dimer model [28], a C-terminal fragment (LEDGF_345–530_) was able to outcompete the LEDGF/p75 dimerization. Surprisingly, LEDGF/p75 dimerization was also interrupted by addition of LEDGF_1–426_ or LEDGF/p52 (Figure 2c), though with different potency. These data reveal that not only the previously indicated C-terminal domain but also a region in the N-terminal part of LEDGF/p75 (aa 1–325) is involved in dimerization. Neither the structured PWWP-domain nor the IBD-domain alone was able to outcompete the dimerization at a protein concentration <1 µM (Figure 2d).

### 3.2. DNA-Induced Dimerization through the N-Terminal Part of LEDGF/p75

Since a strong competition of LEDGF/p75 homodimerization was observed after addition of LEDGF/p52, AlphaScreen experiments were carried out to measure the interaction of LEDGF/p52 with LEDGF/p75 (Figure 3a). These results indicate that the alternative splice variant LEDGF/p52, which lacks the C-terminal part, is able to form dimers. To further narrow down the domain or motif responsible for dimerization, various LEDGF/p75 mutants were tested for their ability to dimerize (Figure 3b). A deletion mutant that lacks both the NLS and the two AT-hook like regions (LEDGF_dNLS+ATh_) was unable to form dimers (Figure 3c). Since it is known that the NLS and AT-hooks are important for DNA-binding and association with chromatin [21], we subsequently purified a DNA-binding-deficient mutant (LEDGF/p75_dDNA_). LEDGF/p75_dDNA_ has five reversed charges on critically important amino acids in the NLS and AT-hooks (R149D, R151D, R182D, K192D, and R196D), and was previously shown to bind DNA with a 30-fold reduced affinity compared to LEDGF/p75 WT [29]. This mutant protein does not dimerize with LEDGF/p75 (Figure 3c). Vice versa, addition of DNA promoted LEDGF/p75 dimerization between the LEDGF/p75 WT and LEDGF/p75_dDNA_ mutant (Appendix A), suggesting that the dimerization of LEDGF/p75 is induced by the presence of DNA.

The amount of DNA in the protein batches was 2 µg/mL or lower (Appendix A). To further confirm this hypothesis, 0.13 U/well of micrococcal nuclease (exo-and endonuclease) was added to the AlphaScreen buffer to digest all remaining DNA in the sample. Nuclease treatment abolished the interaction between two wild-type LEDGF/p75 molecules, supporting the hypothesis that DNA binding is necessary for LEDGF/p75 dimerization (Figure 3c). The dimerization observed for the shorter isoform LEDGF/p52 is also induced by binding to DNA, since both the interaction between LEDGF/p75 with LEDGF/p52 and LEDGF/p52 with LEDGF/p52 was disrupted upon the addition of nuclease (Appendix A).

In cells, LEDGF/p75 tethers several cellular proteins, as well as the HIV-1 integrase, through its IBD domain towards active genes [24,32]. To test whether DNA-driven dimerization affects the interaction with the best-described LEDGF/p75 binding partners, HIV-1 IN and MLL1, the dimerization-deficient mutants were tested in AlphaScreen for their binding abilities. LEDGF/p75_dDNA_ bound to HIV-1 IN (Figure 3d), whereas the binding to the MLL1_1–160_ fragment was abolished (Figure 3d). This differential impact on the interaction with HIV-1 IN or MLL1_1–160_ was confirmed in the AlphaScreen assay with 0.13 U/well of micrococcal nuclease present (Figure 3d). The nuclease removes DNA, and is therefore an alternative method to confirm the results observed with the non-DNA-binding LEDGF/p75 mutant, LEDGF/p75_dDNA_. A similar binding behavior was observed for another cellular binding partner, JPO2, with abolished binding in the absence of DNA (Appendix A). The interaction between LEDGF/p75 and HIV-1 IN remained intact after nuclease treatment (Figure 3d), providing evidence that the LEDGF/p75 interaction with its binding partners can be either DNA-dependent or -independent. Interestingly, HIV-1 IN, which is a pathogen protein binding to LEDGF/p75, may bind independently from DNA to hijack LEDGF/p75 from binding to host binding partners, as proposed earlier [10,24,33].

### 3.3. Detailed Analysis of the Dimerization Domain Located in the C-Terminus of LEDGF/p75

Recently, the structure of the minimal dimerization domain located in the C-terminal part of LEDGF/p75 was resolved by NMR [28]. Based on the obtained data, a model for the interaction was proposed, whereby the IBD forms a domain-swapped dimer with the IBD of another molecule. This complex is stabilized by electrostatic interactions of a negatively charged α-helix (α_6_-helix) formed within the highly flexible C-term (C-term, aa 426–530, Figure 1b). Of interest, several of the amino acids involved in dimerization and located in the IBD are the amino acids that contribute to the formation of two hydrophobic pockets that are important for the interaction with both the cellular binding partners and HIV-1 IN [32]. However, previous data have shown that the residues located at the C-terminus contribute significantly to the dimer stability but do not affect the protein’s capability to interact with cellular partners in vitro [28]. Therefore, LEDGF/p75 may bind as a dimer to its binding partners.

We have further analyzed the dimerization by introducing mutations in full-length LEDGF/p75 and in a LEDGF variant that contains the IBD and C-term, but that lacks the N-terminal part to exclude the dimerization effect induced by DNA binding to the NLS and AT-hook-like regions, LEDGF_325–530_. To study the role of the α_6_-helix within the C-term, we investigated the binding behavior of a fragment that was described before necessary for dimerization that contains the α_6_-helix (LEDGF_325–467_) and a fragment that misses the α_6_-helix, which we would expect to behave as a monomer under the analyzed concentration (LEDGF_325–426_). Both fragments were also tested in the context of the longer protein versions, LEDGF_1–467_ and LEDGF_1–426_ (Figure 4a). Of interest, removing the α_6_-helix did not interrupt the dimerization, but showed a stronger interaction with WT LEDGF/p75, and this both for the short fragments (Figure 4b, LEDGF_325–426_) as for the long fragments (Figure 4c, LEDGF_1–426_).

To study whether the presence of the α_6_-helix within the C-term affects the binding with HIV-1 IN or MLL1_1–160_, the C-terminal truncation mutants were tested in AlphaScreen for their binding abilities. Only the deletion mutant of LEDGF/p75 that completely lacks the C-terminal helix (LEDGF_1–426_) formed a slightly stronger complex with MLL1_1–160_ (Figure 4d), while no difference for binding to HIV-1 IN was observed compared to LEDGF/p75 WT (Figure 4e).

Next, we observed that the introduction of point mutations in the α_6_-helix that abrogate the electrostatic stabilization of the C-term (His_6_-LEDGF_345–530_ E451A, His_6_-LEDGF_345–530_ N454A-K455A or Flag-LEDGF/p75 E451R-E452R) (Figure 5a) did not abort dimerization, but rather reinforced the interaction when testing the binding to WT LEDGF/p75 protein (Figure 5b). In parallel, the His_6_-LEDGF_345–530_ mutants were tested for binding to LEDGF_1–426_, the truncation mutant of LEDGF/p75 missing the complete C-term, including the α_6_-helix. A ~2-fold stronger binding of WT His-LEDGF_345–530_ was observed compared to the mutants His_6_-LEDGF_345–530_ E451A and N454A-K455A (Figure 5c).

The data are summarized in Figure 5d,e, and support the model illustrated in Figure 5d: At protein concentrations <1 µM, LEDGF/p75 can adopt a backfolded autoinhibitory conformation, where the C-term folds back on the IBD forming a “closed” IBD. Introducing reverse-charged mutations in helix 6 results in an “open” conformation rendering the IBD available for dimerization with the IBD and/or C-term of another LEDGF molecule (Figure 5d,e). When testing these point mutants in AlphaScreen for their interaction with HIV-1 IN or MLL1_1–160_, LEDGF/p75 E451R-E452R did not show a difference for binding to HIV-1 IN compared to LEDGF/p75 WT (Figure 5f), while the same mutant generated a higher AlphaScreen signal for interaction with MLL1_1–160_ (Figure 5g). This indicates that in the “open” conformation, when shielding of the IBD by the C-terminus of LEDGF/p75 is absent, the IBD is more available for interaction with MLL1_1–160_. So far, we have shown that the recombinant WT LEDGF/p75 protein forms dimers with a contribution of the N-terminal part (aa 1–325) and the C-terminal part of the protein (aa 325–530). To investigate the role of dimerization for the function of LEDGF/p75 as a molecular tether, we evaluated the impact of dimerization on HIV-1 replication and MLLr-induced leukemia, two pathologies known to be dependent on LEDGF/p75 levels [7,8,10,34].

### 3.4. Dimerization of LEDGF/p75 Is not Required for HIV-1 Replication

HIV-1 integrase hijacks the cellular function of LEDGF/p75 to reach actively transcribed genes in the host genome. HIV-1 replication, and more specifically the viral integration and integration site selection of HIV-1, is hampered in the absence of LEDGF/p75 [7,11,31]. To investigate the effects of dimerization on multiple-round HIV-1 replication, the respective LEDGF/p75 mutants, LEDGF/p75 ATh (which differs from LEDGF/p75_dDNA_ by missing the mutations in the NLS, and only carries the two mutations in the AT-hook like domains), LEDGF/p75 E451R-E452R, and LEDGF_1–441_, were first stably expressed in HeLa P4 cells, previously depleted for endogenous LEDGF/p75 (LEDGF/p75 KD), using lentiviral vectors. Expression levels were evaluated by qPCR and Western blot (Appendix A). The respective cell lines were challenged with HIV-1_NL4.3_ at a MOI of 0.01 and HIV-1 replication was monitored by measuring the p24 protein concentration in the supernatant until day 10, when cells were fully grown, or cytopathic effects were observed (Figure 6). As predicted, HIV-1 replication in HeLa P4 cells depleted for LEDGF/p75 (Figure 6, grey line) was hampered. HeLa P4 cells complemented with WT LEDGF/p75 (Figure 6, brown line) clearly supported HIV-1 replication to much higher levels than noncomplemented HeLa P4 LEDGF KD cells (Figure 6, black line). In addition, complementation with either LEDGF/p75 ATh (Figure 6a), LEDGF/p75 E451R-E452R (Figure 6b), or LEDGF_1–441_ (Figure 6c) genetic variants supported HIV-1 replication similar to cells complemented with WT LEDGF/p75 (Figure 6, brown line).

### 3.5. LEDGF/p75-Dimerization Is Important for LEDGF/p75-Dependent MLL1 Fusion Mediated Leukemia

Depletion of LEDGF/p75 expression is known to impair the clonogenic growth of MLL1-fusion (MLLr)-transformed hematopoietic cells [8]. To evaluate the importance of the newly defined LEDGF/p75 dimerization interfaces, we investigated the colony-forming capacity of MLL1-AF9 leukemic cells upon overexpression or back complementation of LEDGF/p75 wild type, LEDGF/p75 ATh, LEDGF/p75 E451R-E452R, and LEDGF_1–441_. LEDGF/p75_dDNA_ (the recombinant variant of LEDGF/p75 ATh) did not interact with MLL1_1–160_ in vitro (Figure 3), while LEDGF/p75 E451R-E452R and LEDGF_1–426_ displayed a stronger interaction, due to a more “open” conformation of the IBD (Figure 5g).

MLL1-AF9-positive THP1 cells were first stably transduced with a mock lentivector (Mock) or a lentivector expressing wild-type or mutant RNAi-resistant LEDGF/p75. All vectors carried a blasticidin resistance cassette-enabling selection. Overexpression of LEDGF/p75 was verified by Western blot and q-PCR (Appendix A), and cells were subsequently seeded in methylcellulose. Half of the cells selected to overexpress LEDGF/p75 WT or the respective mutants were subsequently transduced with a lentiviral vector to knock down the expression of endogenous LEDGF/p75 (LEDGF/p75 KD), creating cell lines in which expression of exogenous LEDGF/p75 could rescue the depletion of the endogenous protein.

In the absence of endogenous LEDGF/p75 (LEDGF KD), a 3-fold decrease in the number of colonies was observed compared to the control (Figure 7; Control (Mock)) as reported before [32]. This reduction in colony formation was partially restored upon LEDGF/p75 back complementation (Figure 7; LEDGF/p75 WT BC—grey bar). The AT-hook mutant, LEDGF/p75 ATh BC, and LEDGF_1–441_ were able to rescue the depletion of endogenous LEDGF/p75 as well. In striking contrast, the dimerization-deficient LEDGF/p75 E451R-E452R was not able to rescue reduced colony formation (Figure 7; LEDGF/p75 E451R-E452R BC—grey bar) after depleting endogenous LEDGF/p75. Overexpression of this mutant even seemed to interfere with the function of endogenous LEDGF/p75.

These data show that expression of LEDGF_1–441_, and thus, the absence of the C-terminus, restores the transformation potential of THP1 cells to a similar extent as detected for LEDGF/p75 WT. However, the importance of the α_6_-helix is clearly demonstrated by the impaired clonogenic growth upon LEDGF/p75 E451R-E452R expression, hinting towards a structural role of the α_6_-helix or an upstream function impeding colony formation, apart from the MLL1 interaction.

## 4. Discussion

Dimer formation of LEDGF_345–467_ was recently demonstrated by NMR [28]. Resolution to molecular detail revealed that dimer formation is induced by domain swapping within the integrase-binding domain (IBD). The dimer is additionally stabilized by electrostatic interactions with the negatively charged α-helix (α_6_-helix) formed in the intrinsically disordered C-terminal region. Interaction with the cellular binding partners did not lead to dimer dissociation, but only to detachment of the C-terminal region, including the α_6_-helix.

When further unraveling the biochemical details of LEDGF/p75 dimerization, we observed that both DNA binding to the NLS and AT-Hook-like regions and the C-term of LEDGF/p75 play a role in the regulation of the transition of monomer to dimer and vice versa (Figure 8a). Focusing first on the N-terminal part of the protein, we verified in protein–protein binding assays that next to LEDGF/p75, the shorter splice variant LEDGF/p52—which lacks the IBD and C-term—can form dimers as well. The dimerization of LEDGF/p52 and p75 is strongly dependent on the presence of DNA. Moreover, DNA binding is not only regulating dimerization, but also affects the interaction between LEDGF/p75 and one of its cellular binding partners, MLL1. Interestingly, binding of another well-characterized binding partner, HIV-1 integrase, was not dependent on the presence of DNA. Different explanations are possible to understand the nature of the DNA-induced conformational change and its differential impact on the interaction with the IBD for the two tested binding partners. First, binding of LEDGF/p75 to DNA may induce an allosteric conformational change in the protein, leading to a change in the accessibility of the IBD, allowing other cellular proteins to bind. Alternatively, the presence of DNA in the sample may bridge several LEDGF/p75 molecules, bringing them close together and increasing the local concentration of LEDGF/p75. This “crowding” effect may result in a conformational change, increasing the binding potential of the IBD and giving access to cellular binding proteins. Crowding followed by conformational change is supported by recent findings for proteins that form phase-separated condensates, such as Med1, Brd4 [35], HP1α [36], and MeCP2 [37], all proteins with similar functions and/or domains as LEDGF/p75 or interaction partners of LEDGF/p75. When HIV-1 enters the cell, it needs to compete with cellular binding partners for binding to LEDGF/p75. A tight affinity [38] and structurally prearranged binding site on integrase favors the interaction between LEDGF/p75 and HIV-1 IN over the interaction with the cellular binding partners, which contain only an unstructured IBD-binding motif (IBM) [24]. Alternatively, HIV-1 IN may favor the monomeric IBD, whereas MLL1 prefers an IBD-swapped dimer. Binding sites for MLL1 and HIV-1 IN on the IBD are overlapping, but they are not identical [32], and also, the structural organization of both proteins around the IBD-swapped dimer is different [39,40]. Figure 8b,c show in silico that dimeric assembly and the displacement of α_6_-helix are compatible with the binding of MENIN/MLL1, while the bulky HIV-1 IN prevents the formation of the dimer form. The dimeric organization of HIV-1 integrase in its catalytic core [39] can bridge two LEDGF/p75 molecules, and as such may mimic the effects of the dimeric assembly of LEDGF/p75 (Figure 8d). HIV-1 IN interacts, as an essential part of the HIV-1 preintegration complex, with LEDGF/p75 before association with the chromosomal DNA [9,41], whereas MLL1, in turn, may only interact with chromatinized LEDGF/p75. This may hint towards an advantage for integrase, capturing LEDGF/p75 earlier and therefore hijacking LEDGF/p75 before encountering the chromatin and its cellular binding partners.

Next, we focused on the dimerization region, which is located at the C-term of the protein and was recently resolved via NMR of the short LEDGF fragment, LEDGF_345–467_ [28]. According to the model, two LEDGF IBDs can dimerize via a swapped domain in the IBD, with improved stabilization by the additional α_6_-helix in the C-terminal part of LEDGF/p75. However, when investigating the involved amino acid residues of the α_6_-helix in the C-term in vitro in an AlphaScreen assay, we also observed, next to the dimerization, backfolding of the protein when in the monomer form. LEDGF_345–530_ WT shows only limited interaction with LEDGF/p75 WT at low protein concentrations, indicating that the C-term of one protein is folding back on the IBD of the same molecule in *cis* (Figure 5d,e), while the α_6_-helix mutants LEDGF_345–530_ E451A and N454A-K455A show an improved interaction with LEDGF/p75 WT, indicating that the mutants are opening up their IBD for dimerization. This contrast with the NMR model can be explained by a difference in binding dynamics and/or protein concentration. All protein concentrations used in the AlphaScreen assay are in the nanomolar range, while NMR analysis was performed at micromolar concentrations. Our data indicate that at a lower protein concentration, the C-terminus is folding back on its own IBD in favor of the monomer form. Interaction studies with the cellular binding protein MLL1 have revealed that the mutant form, with the “open” IBD conformation, is inducing a stronger interaction (Figure 5g).

Finally, we investigated the potential functional impact of dimerization by analyzing the most interesting mutants in a classical (i) cellular HIV-1 replication assay and (ii) in a colony-forming unit (CFU) assay to measure leukemogenesis by an MLL1 fusion (MLLr). For the HIV-1 infectivity assay, LEDGF/p75 knockdown (KD) cells were generated, with back complementation by LEDGF/p75 WT, LEDGF/p75 ATh, LEDGF/p75 E451R-E452R, and LEDGF_1–441_. Yet, no effect of the mutants was observed, as they could successfully support HIV-1 replication similar to cells complemented with wild type LEDGF/p75. These data are consistent with our previous data suggesting that the interaction with HIV-1 integrase is independent from the dimerization state of LEDGF/p75 (Figure 3d).

When investigating the colony-forming capacity of MLL1-AF9+ leukemic cells upon back complementation of LEDGF/p75-depleted cells with LEDGF/p75 WT, LEDGF/p75 ATh, LEDGF/p75 E451R-E452R, or LEDGF_1–441_, a 3-fold decrease in the number of colonies formed was observed with LEDGF/p75 E451R-E452R compared to cells complemented with wild-type LEDGF/p75. These data indicate that the charges in the α_6_-helix in the C-terminal part of LEDGF/p75 are of great importance for colony formation of MLL1-AF9-positive leukemic cells. In stark contrast, rescue using the LEDGF_1–441_, without the C-term, is possible. We hypothesize, based on all previous data, that binding of MLL1 can only happen when LEDGF/p75 encounters the DNA. At that stage, the C-terminal α_6_-helix is most likely displaced. By mutating the α_6_-helix, we probably perturb flexibility and its opportunity to be expelled by cellular proteins. Another possible explanation would lie more upstream, indicating that the effect we observe in the colony formation assay is not due to impaired binding with MLL1, but rather by another disrupted interaction.

In Figure 8a, we present a model explaining our results and accommodating the swapped dimer of the IBD. DNA binding at the N-terminus of a LEDGF/p75 monomer acts as a trigger to regulate dimerization of the IBD via a swapped dimer. This effect is possibly mediated by DNA-induced allosteric conformational changes or by protein crowding upon DNA binding. This trigger may displace the α_6_-helix in the flexible C-term from binding on its own IBD towards binding with another IBD. Binding of DNA also leads to an increased binding affinity of LEDGF/p75 for its cellular binding partner MLL1, possibly via the IBD dimer, but without the need of the α_6_-helix still being bound to the IBD, as we see in the rescue experiment (Figure 7). Next to MLL1, several other binding partners interact with the LEDGF IBD domain. These binding partners are characterized by an IBD-binding motif (IBM), of which phosphorylation affects IBD affinity [26]. Previous research has shown that binding of cellular partners is mutually exclusive. Since the binding interface between the different binding partners is highly similar, it is likely that binding of these proteins is affected by the conformation of LEDGF/p75. We confirmed that a DNA-induced dimer of LEDGF/p75 is necessary for the binding of JPO2 (Appendix A). Additionally, binding and functional studies in context of MLLr-leukemia indicate that backfolding on the IBD domain prevents MLL1 from binding LEDGF/p75. We hypothesize that other cellular proteins can be prohibited from binding the IBD domain when backfolded, potentially regulating protein function. Through an affinity independent of phosphorylation and a slightly different binding pose to IBD, HIV-1 integrase distinguishes itself from cellular binding partners. Here, this is extended through the observation that integrase binding is independent of the LEDGF-confirmation, and this translates to an unaffected HIV-1 replication cycle (Figure 6). Further understanding of the effect of dimer formation on the interaction with these cellular binding partners is of interest for studying these protein-related diseases, such as medulloblastoma in the case of JPO2 [42,43], MLL1-rearranged leukemia for MLL1 [8,34], Rett Syndrome in the case of MeCP2 [44,45], and autism spectrum disorder in the case of PogZ [24,46]. Additionally, insight into the impact of LEDGF dimerization may help to elucidate the role of LEDGF in several other cancers [15,47,48,49,50] and allergic disorders such as asthma or atopic dermatitis, associated with LEDGF autoantibodies [51,52,53]. For disease phenotypes affected by the backfolding of LEDGF/p75, such as MLL1, securing LEDGF/p75 in a backfolded position with, for example, glue molecules [54] or synthetic antibodies [55] would be a novel treatment approach.

## 5. Conclusions

Our results shed light on a potential mechanism whereby the C-terminal tail can fold back on the IBD of LEDGF/p75, possibly protecting this domain from undesired interactions before docking to the chromatin. Chromatin docking, be it by an allosteric conformational change or by crowding and in an increase in effective concentration, triggers a conformational change of the IBD, and its opportunity to dimerize. The integrase protein of HIV-1 evolved to circumvent this system so that it can interact with the IBD of LEDGF/p75 in the absence of chromatin.

## Figures and Tables

**Figure 2 cells-13-00227-f002:**
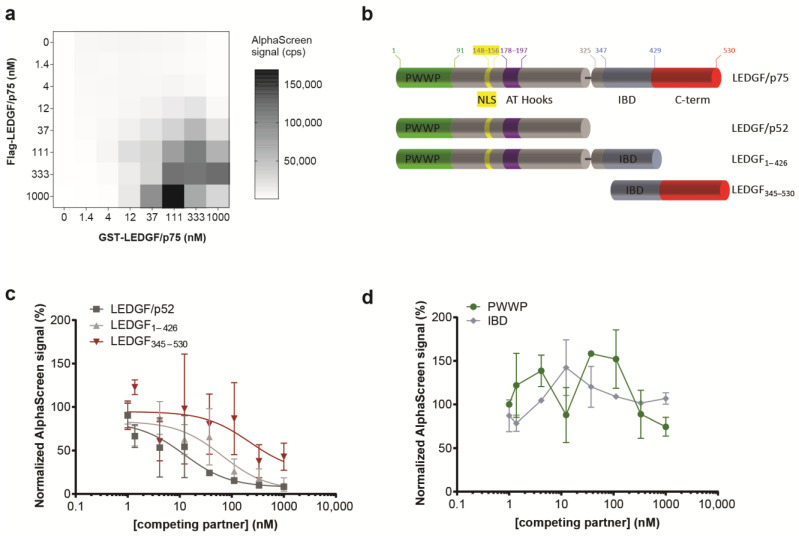
Identification of LEDGF/p75 domains associated with dimerization. (**a**) Cross-titration between Flag-LEDGF/p75 and GST-LEDGF/p75 confirms the dimerization. Data points for cross-titrations are in singlet. (**b**) Schematic representation of the various deletion constructs generated to pinpoint the dimerization domain. (**c**,**d**) The interaction between 150 nM GST-LEDGF/p75 and 150 nM Flag-LEDGF/p75 was outcompeted by increasing concentrations of (**c**) His_6_-LEDGF/p52, His_6_-LEDGF_1–426_, or LEDGF_345–530_ in an AlphaScreen assay but not by (**d**) the PWWP domain or the IBD domain. Error bars in all AlphaScreens represent s.d. calculated from three independent experiments. IC_50_s were calculated when possible. His_6_-tagged LEDGF/p52 outcompeted the interaction of Flag-tagged LEDGF/p75 and GST-tagged LEDGF/p75 with an apparent IC_50_ of 12.2 nM [95% confidence interval (CI), 0.6921 to 67.68]. For LEDGF_1–426_, the IC_50_ increased almost 5 times (IC_50_, 65.5 nM; CI, 18.82 to 226.3), while in the presence of LEDGF_345–530,_ the IC_50_ increased almost 20 times (IC_50_, 222 nM; CI, indefinite).

**Figure 3 cells-13-00227-f003:**
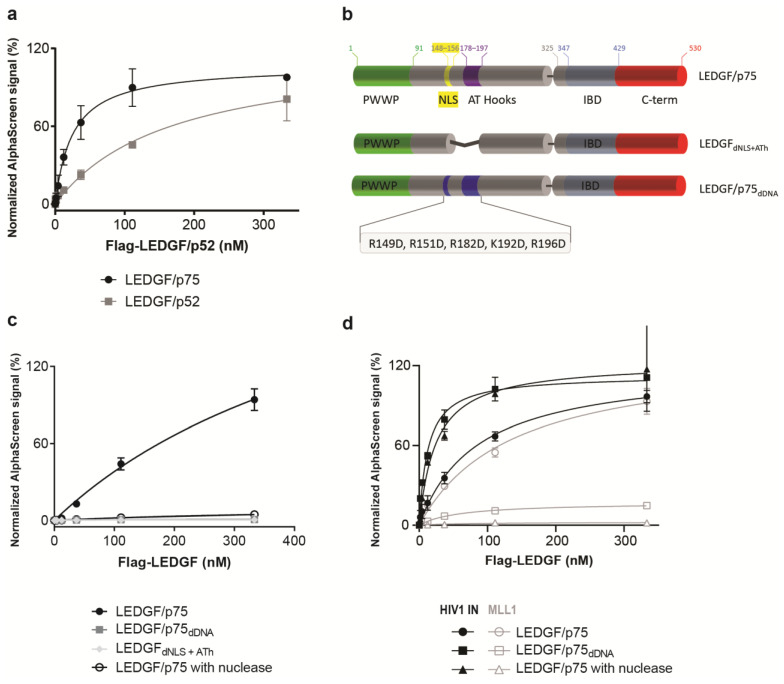
DNA-dependent dimerization of the N-terminal part of LEDGF/p75. (**a**) Confirmation of the dimerization between varying concentrations of LEDGF/p52, which lacks the IBD, and 100 nM LEDGF/p52 or LEDGF/p75. (**b**) Schematic representation of the various constructs generated to pinpoint the dimerization domain in the N-terminal part are indicated. LEDGF_dNLS+ATh_ lacks the NLS and AT-hooks. In LEDGF/p75_dDNA_, five positively charged amino acids were replaced by negatively charged amino acids to interfere with the DNA interaction [29]. (**c**) AlphaScreen assay with 111 nM GST-tagged LEDGF/p75 and varying concentrations of Flag-tagged LEDGF_dNLS+ATh_ (grey diamond), LEDGF/p75_dDNA_ (grey square) or LEDGF/p75 WT with addition of 0.13 U/well micrococcal nuclease to the AlphaScreen buffer (black, open circle). All three conditions abolished the dimerization. (**d**) Interaction of LEDGF/p75 WT (black circles) or LEDGF/p75_dDNA_ (black squares) with 150 nM His_6_-tagged HIV-1 integrase in an AlphaScreen assay. Addition of 0.13 U/well micrococcal nuclease in the buffer (black triangle) did not hamper the interaction between LEDGF/p75 WT and HIV-1 integrase. The interaction between LEDGF/p75 WT (empty circles) or LEDGF/p75_dDNA_ (empty squares) with 350 nM MLL1_1–160_-GST is shown as well. Addition of 0.13 U/well micrococcal nuclease in the buffer (empty triangle) abolished the interaction between LEDGF/p75 WT and MLL1. Error bars in all AlphaScreen experiments represent standard deviation calculated from at least two independent experiments.

**Figure 4 cells-13-00227-f004:**
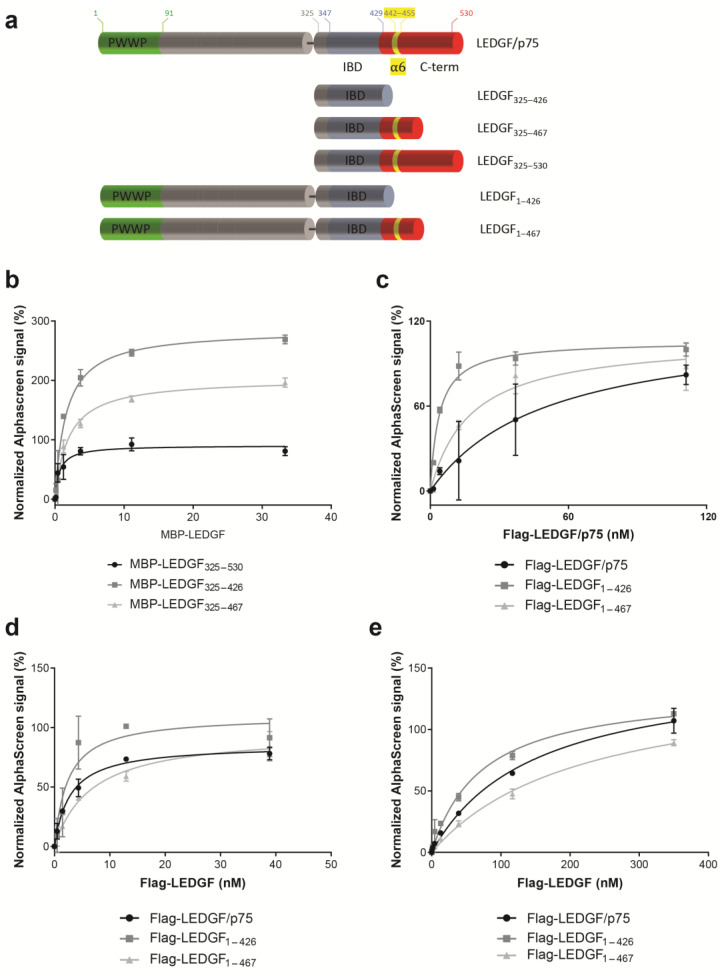
Narrowing down the minimal, stable C-terminal dimerization domain. (**a**) Various LEDGF/p75 deletion constructs were generated to investigate the role of the C-terminus in dimerization. (**b**) AlphaScreen assay with 100 nM of His_6_-LEDGF_345–530_ and various MBP-LEDGF/p75 C-terminal truncations to study the impact of the α_6_-helix on dimerization. (**c**) AlphaScreen assay with 100 nM of full-length GST-tagged LEDGF/p75 and various Flag-tagged LEDGF/p75 constructs to study the impact of the α_6_-helix on dimerization. (**d**) AlphaScreen assay to measure dimerization between various Flag-LEDGF/p75 constructs and 150 nM of MLL1_1–160_-GST. (**e**) AlphaScreen assay to measure dimerization between various Flag-LEDGF/p75 constructs and 111 nM of His_6_-tagged HIV-1 integrase. Error bars in all AlphaScreens represent standard deviation calculated from at least two independent experiments.

**Figure 5 cells-13-00227-f005:**
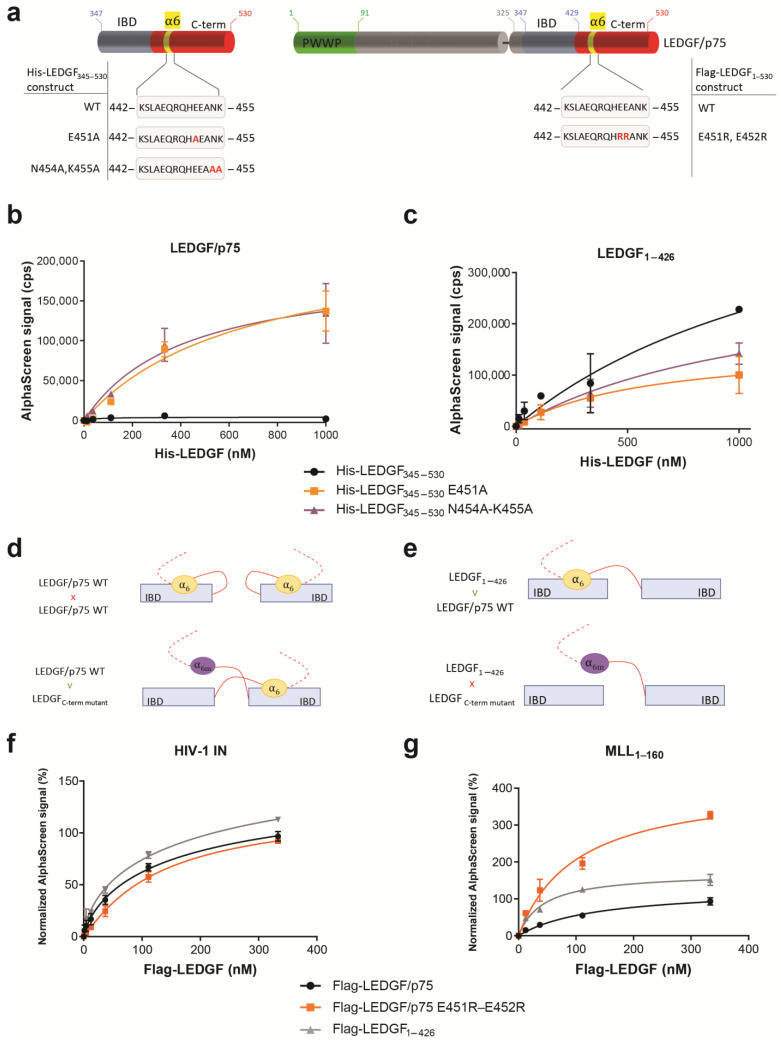
The α_6_-helix contributes to LEDGF/p75 monomer and dimer arrangement. (**a**) Schematic representation of various deletion mutants generated to investigate the effect of the electrostatic stabilization of α_6_-helix in the dimerization interaction. (**b**) AlphaScreen assay of Flag-LEDGF_1–530_ and (**c**) Flag-LEDGF_1–426_ titrated against His_6_-LEDGF_345–530_ (WT, black circle) or the α_6_-helix point mutants destabilizing the electrostatic interaction with the IBD, His_6_-LEDGF_345–530_ E451A (orange squares), or N454A-K455A (purple triangle). (**d**,**e**) Cartoon representation of a LEDGF_345–530_ monomer, with backfolding of the α_6_-helix, and the dimer with a domain swap of the α_6_-helix (IBD swap not indicated in this cartoon). (**f**) Effect of the electrostatic stabilization by α_6_-helix in the interaction between 150 nM His_6_-tagged HIV-1 integrase with Flag-tagged LEDGF/p75 WT (black circles), LEDGF_1–426_ (grey triangles) or LEDGF/p75 E451R-E452R (orange squares). (**g**) Effect of the electrostatic stabilization by α_6_-helix in the interaction between 111 nM MLL1_1–160_-GST with Flag-tagged LEDGF/p75 WT (black circles), LEDGF_1–426_ (grey triangle) or LEDGF/p75 E451R-E452R (orange squares). Error bars in all AlphaScreens represent standard deviation calculated from at least two independent experiments.

**Figure 6 cells-13-00227-f006:**
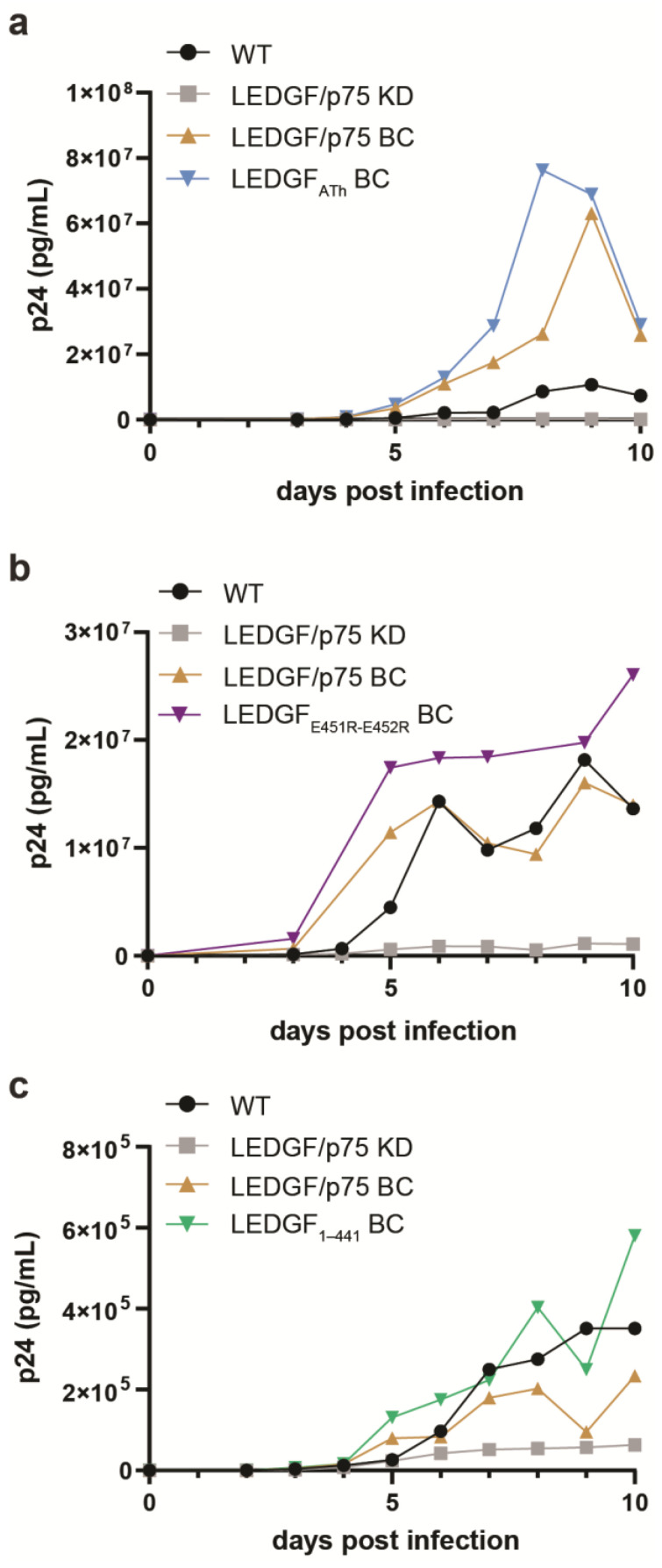
HIV-1 infectivity is not influenced by LEDGF dimerization. Infection of wild-type (WT) HeLa P4 (black), HeLa P4 with a LEDGF/p75-specific knockdown (KD, grey), and back complemented with either WT LEDGF/p75 (LEDGF/p75 BC, brown), (**a**) LEDGF_ATh_ (LEDGF/p75 with R149D, R151D), (**b**) LEDGF_E451R-E452R_, or (**c**) LEDGF_1–441_. As expected, no HIV-1 replication was observed for LEDGF/p75 KD cells. However, HIV-1 replication in back complemented cell lines was similar or better than the WT HeLa P4 cell line. All cells were infected with HIV-1_NL4.3_ (600,000 pg of p24 per condition). Viral replication was monitored by quantifying the p24 amount in the supernatant (using ELISA; Innogenetics, Ghent, Belgium). Experiments were performed in duplicate and repeated twice. A representative experiment is shown.

**Figure 7 cells-13-00227-f007:**
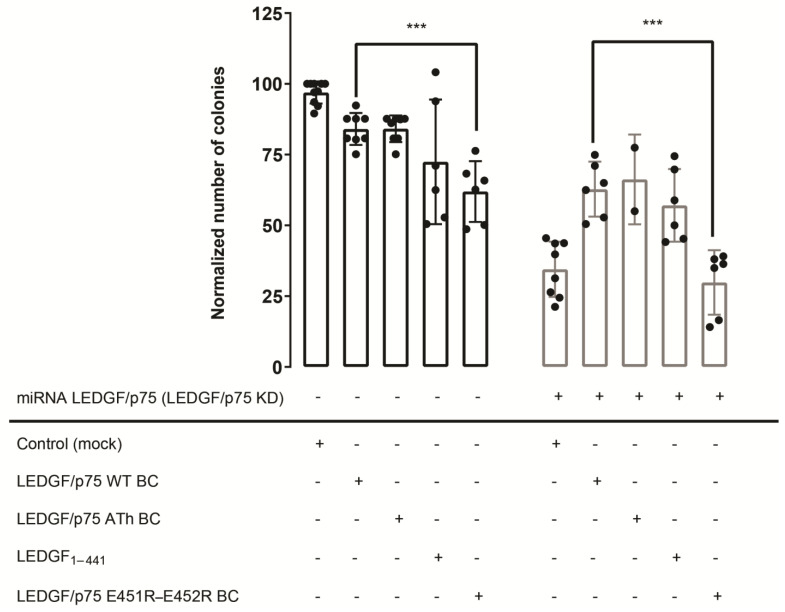
Colony-forming activity of THP1 cells expressing mutant forms of LEDGF/p75 involved in dimerization. THP1 cells were first transduced with a vector for LEDGF/p75 WT BC, LEDGF/p75 ATh BC, LEDGF/p75 E451R-E452R BC, LEDGF_1–441_, overexpressing the respective LEDGF/p75 protein. As a control, wild-type THP1 cells were transduced with a mock vector, containing the same resistance cassette (black bars). After selection, these cells were further transduced with a lentiviral vector encoding WT miRNA-resistant LEDGF/p75 (LEDGF/p75 knockdown), generating cells with back complementation of the respective LEDGF/p75 protein (grey bars). Error bars represent the standard deviation of at least two independent experiments, performed in duplicate. *p*-values are obtained from an unpaired, two-tailed *t*-test. *** *p* < 0.0005.

**Figure 8 cells-13-00227-f008:**
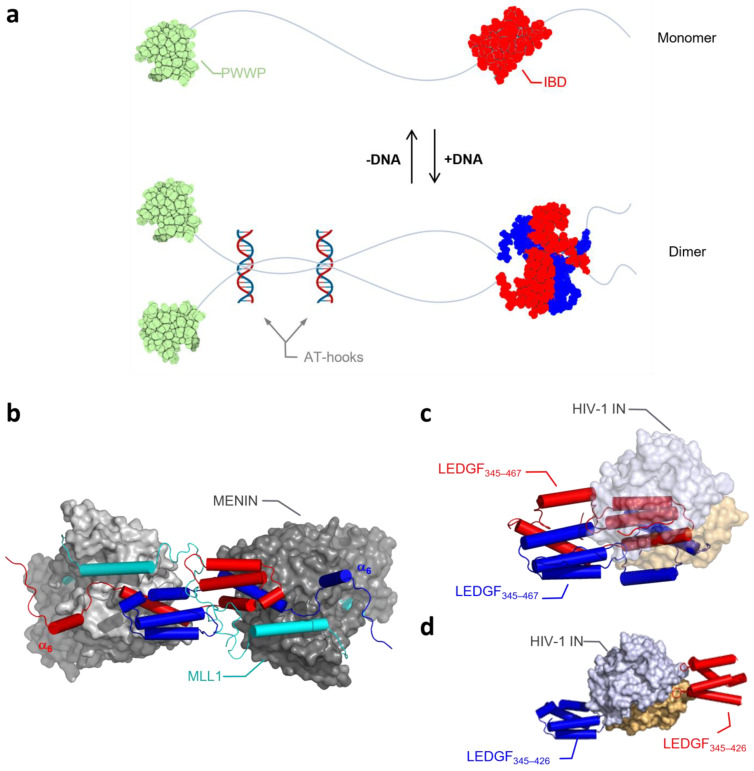
Schematic representation of a DNA-dependent LEDGF/p75 monomer and dimer equilibrium. (**a**) In the absence of DNA at protein concentrations below 1 mM, LEDGF/p75 behaves as a monomer, while DNA binding to the AT-hooks region leads to dimerization through domain swapping and electrostatic stapling of the IBD and C-terminal tail, previously described by Lux et al. [28]). The scheme was created in Biorender. (**b**) The dimeric assembly is compatible with the binding of MENIN/MLL1. MLL1 binding is linked to displacement of α_6_-helix. The model of the ternary com-plex was assembled using the X-ray structure of the IBD/MENIN/MLL1 complex (PDB: 3U88 [40]). (**c**) The dimeric interface is blocked by HIV-1 IN binding. (**d**) HIV-1 IN dimer can bridge two LEDGF/p75 IBD domains, and as such mimic the dimeric arrangement (PDB: 2B4J [39]). Figures in (**b**–**d**) were generated using PyMOL 2.3.

## Data Availability

GraphPad Prism version 7.0 for Windows (Graphpad Software, San Diego, CA, USA) is used for data analysis and graph building. Additional data generated in this study are available from the corresponding author Zeger Debyser (Zeger.Debyser@kuleuven.be).

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
