# Peer review of "The Impact of Lens Epithelium-Derived Growth Factor p75 Dimerization on Its Tethering Function"

_cells, 2024, doi:10.3390/cells13030227_

Round 1
Reviewer 1 Report
Comments and Suggestions for Authors
The growth factor LEDGF/p75 is a large protein that has been studied for many years by some members of the scientific community, including the authors of this article. Its roles as a transcriptional co-activator or cellular cell stress survival factor are known, but this protein is being extensively studied because of its involvement in many diseases such as HIV infection or leukemia. Its architecture in structural and functional domains is already well documented. Continuing to study this protein is of great importance. In particular, the authors in this paper provide the first evidence of the role of DNA in the dimerization dynamics of LEDGF/p75, which may or may not affect its interaction with the proteins involved in the two diseases chosen (to begin with) by the authors.
A new highthrouput interaction measurement technique was chosen, Perkin Elmer's AlphaScreen, which enabled the authors to access the quantities of protein required to achieve dimerization, and to describe the protein regions impacted, using a large number of deleted or mutant constructs. This work is comprehensive and serious, and opens up good avenues for functional hypotheses.
In vivo experiments in 2 systems controlled by the authors allow us to trace the impact of these regions on the infection phenotypes. Conclusions differed depending on the system, showing the great adaptability of LEDGF/p75 to different types of infection.
This work follows on from the determination of the dimeric structure of the IBD domain of LEDGF/p75, whose stability as a function of protein concentration is questioned by the authors, in line with the necessity of the dynamics of monomer-dimer exchange for the hacking focus of viral proteins to facilitate integration of the viral genome into the chromosome.
An interesting allusion to a "crowding" effect or the formation of phase-separated condensates is suggested in the discussion. This seems to be a good system to explore these concepts.
In conclusion, we realize that this system is very complicated, multifactorial and dynamic. This is reflected in the heaviness of the introduction and the length of the discussion, which could perhaps be rewritten to be easier to read, without obligation. The new data from this work will advance understanding of the system and should help subsequent research.
Minor corrections :
Duplicate the graph in figure 3d to separate the curves belonging to HIV and MLL, or add a legend to differentiate them.
Please specify in the legend to figure 8 b-c-d that the structures shown are in silico models (specify the software? AlphaFold?).
Author Response
We thank the referee for the positive feedback and suggestions. We revised our manuscript addressing the comments of the reviewer. In the attached file, we address point-by-point the comments of the reviewer and explain which sections were altered or included in the new version of the manuscript (bold, blue text).

Reviewer 2 Report
Comments and Suggestions for Authors
In the current manuscript, Brouns et al. have tried to define DNA-induced dimerization dynamics in LEDGF/p75 that influence its interactions as a molecular tether in pathways relevant to leukemia and HIV integration. The structural elucidation sheds light on how DNA interactions and mutations might alter LEDGF dimerization to modulate protein binding partners. It’s an interesting study but need some further work/clarifications before considered to publication.
1. Expand on the rationale for focusing on MLL1 and HIV-1 integrase interactions. Elaborate on why these two binding partners were prioritized and how well they generalize to other cellular interactions.
2. Expand on the rationale for focusing on MLL1 and HIV-1 integrase interactions. Elaborate on why these two binding partners were prioritized and how well they generalize to other cellular interactions.
3. In this specific study the authors have not identified the species formed by LEDGF/p75 are at all dimers or could be multimeric species.
4. Use the Fig S1 with the error bars in the main figure 2. Fit the Fig 2C to get a comparative value.
5. Data representation in fig 6 could be improved for clarity. Why start measuring the p24 concentration at 3rd day? The starting concentration of p24 is very different at the starting point in different conditions, so it is hard to compare.
6. The authors showed that, Dimerization of LEDGF/p75 is not required for HIV-1 replication. But how LEDGF/p75 effects HIV-1 replication is not clear from their study.
7. Fig 3d, is DNA binding inhibiting the HIVI interactions?
8. Elaborate on the broader impacts and future directions:
a. Beyond MLL1 and HIV-1 integrase, relate structural effects to other cellular interactions.
b. Discuss how these molecular insights could inform drug design or disease treatments.
9. Line 330, It should be ‘DNA-binding deficient’ not, ‘DNA-deficient binding mutant’.
10. Fig 5f, FLAG-LEDGF 1-426 missing.
Author Response
We thank the referee for the constructive feedback and suggestions that helped to improve the manuscript. We revised our manuscript and performed additional experiments addressing the comments of the reviewer. In the attached file, we address point-by-point the comments of the reviewer and explain which sections were altered or included in the new version of the manuscript (bold, blue text).

Reviewer 3 Report
Comments and Suggestions for Authors
The manuscript by Brouns et al. describes interaction experiments with the protein LEDGF/p75 and its variants using biochemical and cellular approaches. I have difficulty evaluating the biochemical part, as the characterization of the proteins used in the assays is not described or shown. It seems that the prep contains DNA traces that already cause dimerization of the proteins? The rest of the results are not clear. A quantification of the purity and amount of DNA will help the reader to assess the relevance of the results. An image of an SDS-PAGE showing the proteins will help. As this part seems to be the main probe of the authors to show the effect of dimerization (5 of the 7 figures showing results). Figure 8 is full of speculations.
Author Response
We thank the reviewer for his feedback. We revised our manuscript and performed additional experiments. In the attached file, we address point-by-point the comments of the reviewer and explain which sections were altered or included in the new version of the manuscript (bold, blue text).

Round 2
Reviewer 2 Report
Comments and Suggestions for Authors
The authors have addressed the concerns adequately. The paper can be published.
Reviewer 3 Report
Comments and Suggestions for Authors
The authors have made a good effort to address the content of DNA in the protein samples and to show the purity of the proteins. In addition, they have incorporated the suggestions of the other reviewers, thereby improving the clarity and significance of the study. The manuscript has been improved sufficiently to warrant publication.
Just a pair of typos:
‘2 µg of each protein was loaded’ should be ‘2 µg of each protein were loaded’ in the legend of Supporting Figure 1.
‘Micorcoccal Nuclease’ should be ‘Micrococcal Nuclease’ in the footnote of Supporting table 2.